# Exploring the influence of different habitats and their volatile chemistry in modulating sand fly population structure in a leishmaniasis endemic foci, Kenya

Iman B. Hassaballa[1,2], Baldwyn Torto[1,2], Catherine L. Sole[2], David P. Tchouassi[1]*

**1** International Centre of Insect Physiology and Ecology, Nairobi, Kenya, **2** Department of Zoology and Entomology, University of Pretoria, Pretoria, South Africa

* dtchouassi@icipe.org

**Data Availability Statement:** All relevant data are within the manuscript and its Supporting Information files.

## Abstract

Phlebotomine sand flies transmit many viral protozoan and bacterial pathogens of public health importance. Knowledge of the ecologic factors influencing their distribution at local scale can provide insights into disease epidemiology and avenues for targeted control. Animal sheds, termite mounds and houses are important peri-domestic and domestic habitats utilized by different sand flies as resting or breeding habitats. However, our knowledge for selection of these habitats by sand flies remains poor. Here, we tested the hypothesis that these habitat types harbor different composition of sand fly species and differ in their volatile chemistry that could influence sand fly selection. To achieve this, we employed CDC light traps following a cross-sectional survey to investigate the distribution of sand flies in the three habitats in an endemic site for leishmaniasis in Kenya. The study was carried out during the dry season, when sand flies are optimally abundant in 2018 and 2020. Sand fly abundance did not vary between the habitats, but species-specific differences in abundance was evident. Measures of sand fly community structure (Shannon diversity and richness) were highest in animal shed, followed by termite mound and lowest inside human dwelling (house). This finding indicates broader attraction of both sexes of sand flies and females of varying physiological states to animal sheds potentially used as breeding or resting sites, but also as a signal for host presence for a blood meal. Furthermore, gas chromatography-mass spectrometric analysis of volatiles collected from represented substrates associated with these habitats *viz*: human foot odor on worn socks (houses indoors), cow dung (animal sheds) and termite mounds (enclosed vent), revealed a total of 47 volatile organic compounds. Of these, 26, 35 and 16 were detected in human socks, cow dung and enclosed termite vent, respectively. Of these volatiles, 1-octen-3-ol, 6-methyl-5-hepten-2-one, α-pinene, benzyl alcohol, *m*-cresol, *p*-cresol and decanal, previously known as attractants for sandflies and other blood-feeding insects, were common to the habitats. Our results suggest that habitat volatiles may contribute to the composition of sand flies and highlight their potential for use in monitoring sand fly populations.

**Funding:** The support through a PhD scholarship to IBH by the German Academic Exchange Service (DAAD) (Grant number 91672086) through the African Regional Postgraduate Programme in Insect Science (ARPPIS) tenable at *icipe* is greatly acknowledged. This study was partly supported by the project, Combatting Arthropod Pests for better Health, Food and Climate Resilience (Project number: RAF-3058 KEN-18/0005) funded by Norwegian Agency for Development Cooperation (Norad). We also acknowledge the financial support for this research by the following organizations and agencies: UK's Foreign, Commonwealth & Development Office (FCDO), the Swedish International Development Cooperation Agency (SIDA), the Swiss Agency for Development and Cooperation (SDC), the Federal Democratic Republic of Ethiopia and the Government of the Republic of Kenya. The views expressed herein do not necessarily reflect the official opinion of the donors. The funders had no role in study design, data collection and analysis, decision to publish, or preparation of the manuscript.

**Competing interests:** The authors have declared that no competing interests exist.

## Author summary

Understanding the ecology of sand flies is critical to developing control measures. An important ecologic adaptation of sand flies is their selective use of habitats as resting or breeding habitats. However, the basis for this preference is not fully understood. Here, we sought to understand the distribution of sand flies in three different habitats namely animal shed, human dwelling (houses indoors) and termite mound, and analyzed the chemical cues associated with these habitats. The study was conducted in Rabai village in Baringo County, Kenya, endemic for visceral and cutaneous leishmaniasis. By analyzing sand flies surveyed at different time points during the dry season, we found that sand fly abundance and diversity varied between the habitats. We collected and analyzed volatile organic compounds from represented substrates associated with these habitats and found commonality in some compounds previously known as attractants for sand flies across the habitats. These volatiles may contribute to the composition of sand flies in these habitats that can potentially be exploited in sand fly surveillance and control purposes.

## Introduction

Phlebotomine sand flies are dipterans belonging to the family Psychodidae. They transmit many viral, protozoan and bacterial pathogens of public health importance. Notably, is their vectoring role of leishmaniasis, a neglected tropical disease which causes substantial morbidity and mortality in many parts of the world. Leishmaniasis has three clinical forms, namely, visceral leishmaniasis (VL), cutaneous leishmaniasis (CL) and mucocutaneous leishmaniasis [1,2]. The disease is prevalent in much of eastern Africa including Kenya where VL and CL occur [2,3]. Recent reports have shown that the geographic range of VL in Kenya is expanding, with ~900 new cases annually [4] and case fatality rate of up to 7% in an outbreak situation [4,5]. Repeated outbreaks of VL in previously non-endemic areas have been reported in northern Kenya [6]. Most recent CL epidemics in Kenya were reported in Gilgil, Nakuru County in the Rift Valley area [7,8].

Control of leishmaniasis relies largely on chemotherapy to treat infected humans. However, the efficacy of this option is variable as the drugs used are toxic and expensive including reports of parasite resistance to the available drugs [9,10]. Licensed vaccines against leishmaniasis are non-existent [11]. Insecticides are a major tool for vector-borne disease control including leishmaniasis targeting sand flies [12,13]. Yet, in many foci, sand fly control is often a byproduct of anti-malarial vector control [14]. In Kenya, leishmaniasis control largely focuses on case management with minimal sand fly control mainly in response to outbreaks. The increasing public health impact of leishmaniasis calls for the need to develop integrated vector management (IVM) strategies for prevention and control of sand flies.

New ways of controlling leishmaniasis can be achieved through improved understanding of sand fly behavioral ecology especially resting and breeding habitats. Control of sand flies has been attempted using insecticides, environmental management or other biocontrol agents (e.g., *Metarhizium anisopliae*) targeting peri-domestic and non-domestic resting/breeding habitats [12,13]. The resting ecology is complex as sand flies utilize a variety of habitats including cracks and crevices, caves, tree holes, rock walls, a stonewall, termite mounds, human habitations, animal sheds, animal burrows and Acacia species [15,16]. However, preference for these ecological sites exists among different vector species and has been exploited in targeted control. For instance, there have been attempts to control *Phlebotomus orientalis* in Sudan

through experimental research by spraying Acacia woodlands with insecticides because of its strong association with this plant [16,17]. Previous reports noted the importance of house spraying with residual insecticides where vectors were domestic [12–14]. Other examples include spraying insecticides on large tree trunks, the preferential resting site to experimentally control *Lutzomyia* in Neotropical forests [18], rodent burrows used as resting and breeding sites for *Phlebotomus papatasi*, and termite mounds preferentially utilized by *Phlebotomus martini* as resting/breeding sites [9,19]. Environmental management targeting suitable habitats in Brazil has been shown to impact sand fly abundance and *Leishmania* infection rates [20,21]. The epidemiologic significance calls for improved understanding about variation in this behavioral adaptation among sand fly species.

In Kenya, Baringo County in the Rift Valley, is the only known leishmaniasis focus where both VL and CL forms of the disease co-occur [2]. The vectors of VL and CL are *P. martini* Parrot and *Phlebotous duboscqi* Neveu-Lemaire, respectively, in this focus, which has allowed for comparative studies of their bio-ecology in relation to leishmaniasis transmission dynamics. Previous studies in this locality observed that human habitations, termite mounds and animal sheds were among resting habitats of epidemiologic importance for these sand fly species [22–25]. Differential sand fly composition and abundance patterns as well as seasonality in these resting sites have also been recorded [22,24], although the basis for this trend is poorly understood.

In this study, we tested the hypothesis that the composition of sand flies varies in different habitats and that the habitats differ in their volatile chemistry that could influence sand fly selection. This concept is well founded following evidence of attraction of hematophagous flies including sand flies to volatile sources including animal skin [26–28] and host metabolites such as faeces [29], urine [30], as well as habitats [31]. To achieve this, we investigated two specific objectives: i) to assess and compare the sand fly abundance and diversity patterns in the three selected habitat types (houses indoors, termite mound, animal shed), and ii) explore the volatile organic compounds associated with these habitats.

## Materials and methods

### Ethics statement

Approval for the study was sought from the Scientific Ethics and Review Committee of the Kenya Medical Research Institute (SERU-KEMRI) (Protocol number: 3312). Additionally, verbal consent was obtained from the chief of Rabai village and heads of households selected for sampling sand flies from houses indoors and outdoors.

### Study site

Field survey of sand flies was conducted in Rabai village (0.45866 N, 35.9889 E) located near Marigat town, Baringo County (Fig 1). Marigat sub-county is a semi-arid ecology situated at *ca*.1000 m above sea level and ~ 250 km north-west of Nairobi, Kenya (Fig 1). Mean annual rainfall in the area is around 300–700 mm, with nightly temperatures of 16˚C and maximum daily temperatures of 42˚C [32]. The vegetation types include scattered Acacia trees, Cactus plants, *Balanites* spp, *Prosopis juliflora* trees and *Commiphora* bushes. Rabai is endemic for CL and VL and known to harbor both *P. duboscqi* and *P. martini* among a host of other *Sergentomyia* species [9]. Houses in Rabai are commonly of two types: corrugated zinc/iron or mud wall with thatched or corrugated zinc roof. Over 60% of the inhabitants own livestock mainly cows, sheep, chicken and goats (being predominant). In addition, there are numerous termite mounds interspersed throughout the landscape [24]. Animal sheds are generally located outdoors and close to houses.

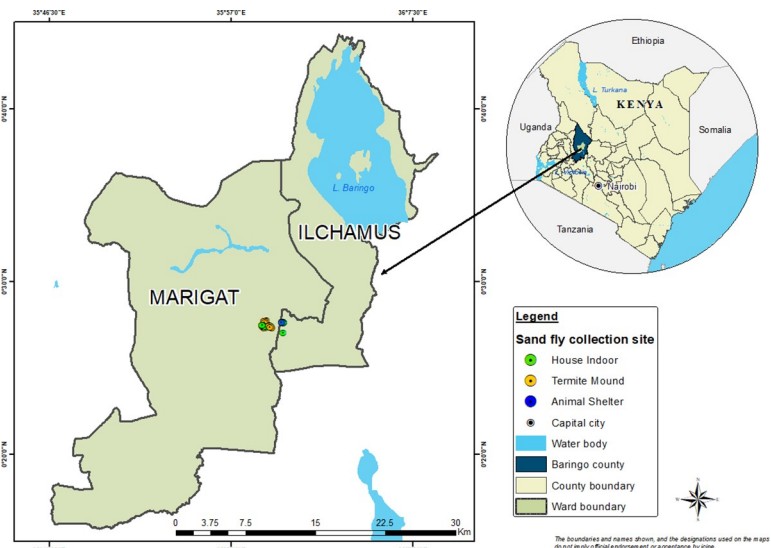

**Fig 1. Map of Marigat sub-county, Baringo County, Kenya showing the study site.** The map was designed using ArcMap 10.2.2. With the ocean and lakes base layer derived from Natural Earth (http://www.naturalearthdata.com/, a free GIS data source). The sample points were collected using a handheld GPS device (Garmin etrex 20), and the county boundaries for Kenya derived from Africa Open data (https://africaopendata.org/dataset/kenya-counties-shapefile, license Creative Commons).

## Sand fly sampling and processing

We employed a cross sectional survey of sand flies conducted in December 2018 and January 2020 during the dry season. Adult phlebotomine sand flies were sampled using CDC light traps (Model 512, John W. Hock Company, Gainesville, Florida, USA). This trapping tool has been shown to effectively target *Phlebotomus* and diverse *Sergentomyia* species [24]. Between 3–4 traps were deployed daily in each habitat (animal shed, houses indoors and termite mound) for eight consecutive nights. Only one trap was set up in each habitat type at a given time. Trapping in new habitats were targeted daily. Traps were positioned at 50–100 m intervals along a transect (indoor households, and outdoor in the termite mound and animal shed) with each trap position georeferenced using a handheld Global Positioning System (GPS) device (Garmin etrex 20). Traps were set at a height of approximately 30–50 cm above the ground (Fig 2). Traps were set up around 18:00 h and retrieved at 06:00 h the following day. The number of trap nights were 36, 10 and 34 for termite mounds, animal sheds and houses indoors, respectively, in December 2018. Trap nights in January 2020 were 22, 14 and 24 for termite mounds, animal shed and houses indoors, respectively. Captured sand flies were knocked down using triethylamine for sorting. Thereafter, they were stored in liquid nitrogen and transported to the laboratory at the International Centre of Insect Physiology and Ecology (*icipe*), Nairobi for storage at −80˚C until further processing.

## Sand fly species identification

For each sand fly specimen, the head and genitalia were excised and mounted on a slide and cover slip using Berlese's medium. One day after allowing the slides to dry, species level identification was achieved by microscopically observing cibarial armatures (*Phlebotomus* or *Sergentomyia*), male genitalia or female spermathecae and pharynx using established morphological keys [33,34].

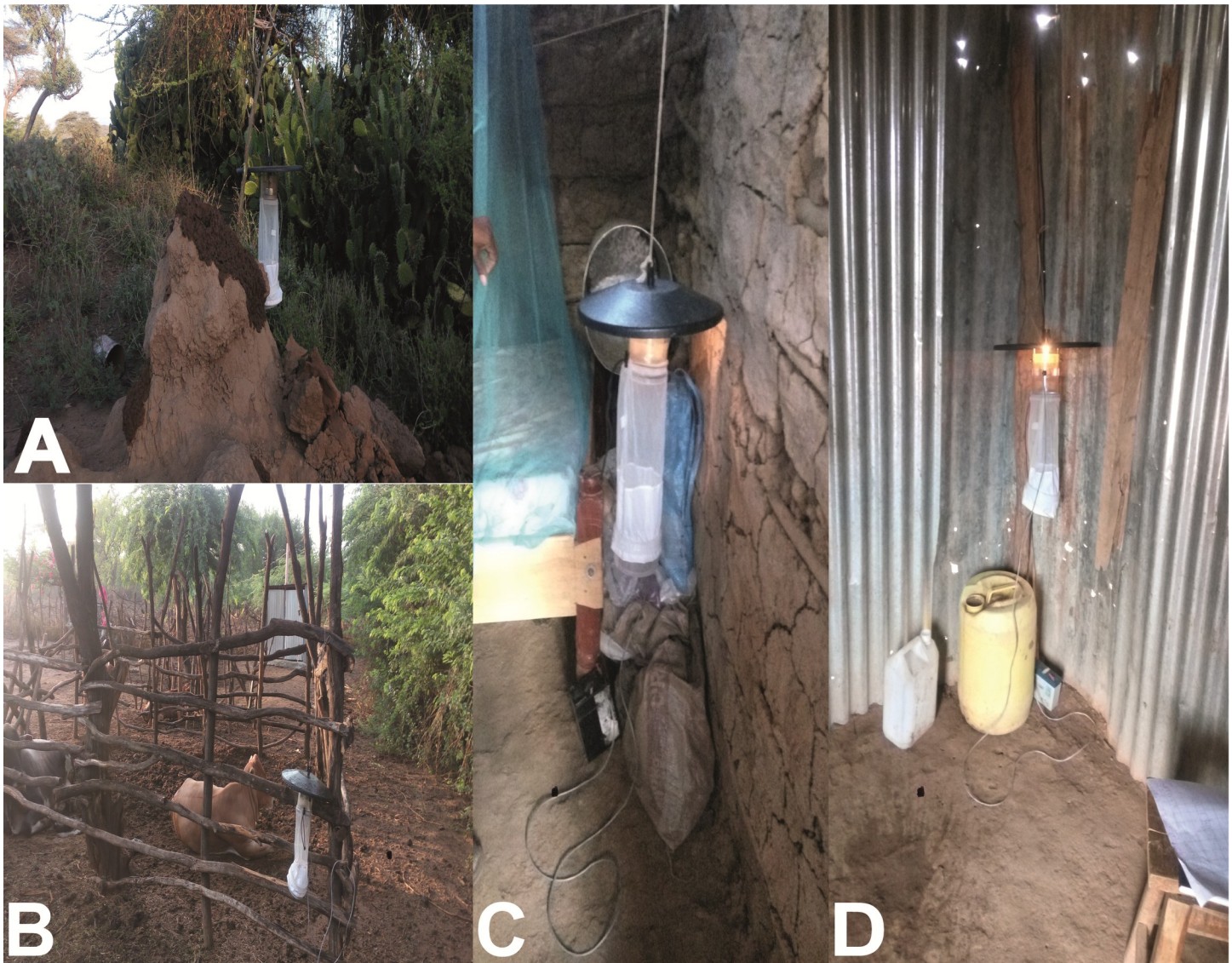

**Fig 2. Sand fly sampling habitats in Rabai village with CDC-LT A) on a termite mound, B) near an animal shed, C) indoors in a mud-type house, D) indoors in a corrugated zinc-type house.** (Source: Iman B. Hassaballa).

## Collection of volatiles

Odors were collected from three substrates representative of the three habitat types: fresh cow dung (animal shed), human foot odors on worn socks (houses indoors), and termite mound. Odor collection was conducted concurrently during sand fly trapping in January 2020. Fresh cow dung (600 g) and human worn socks (a pair) were placed in quick-fit airtight glass chambers (1600 mL), whereas volatiles were collected directly from the termite mound in the field (Fig 3). To obtain human foot odors, male volunteers (aged 35–60 years old) wore a pair of socks (made of cotton, Kaite Socks, China, solvent and oven-cleaned) overnight for 12 h. The volunteers were asked not to use any soap when showering, nor apply lotion or perfume 24 h prior to sampling. The human volunteers and dung were from represented houses indoors and animal sheds, where sand flies were trapped. Odors from selected termite mounds (where sand fly trapping occurred) were collected by enclosing the vent in an airtight oven bag

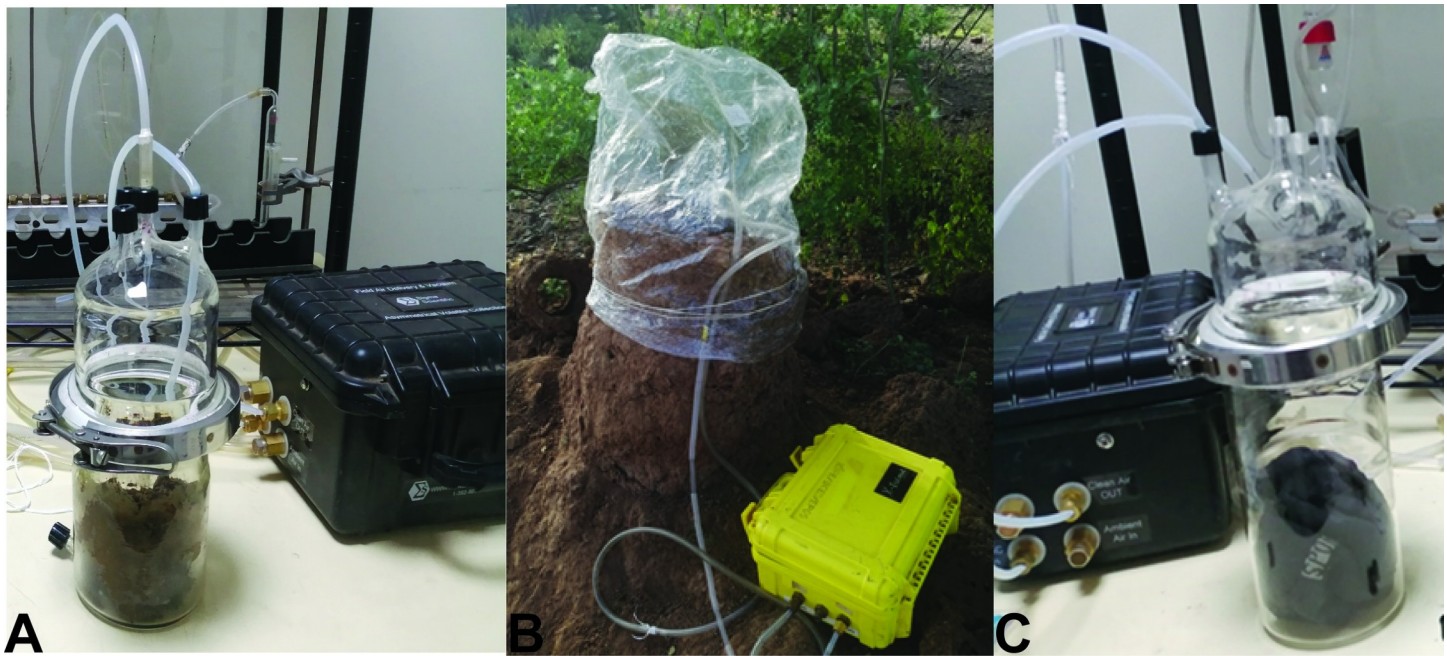

**Fig 3. Collection of volatiles from three substrates representing the different habitats in Rabai village, Marigat sub-county, Kenya A) cow dung (animal shed), B) termite mound (termite vent) and C) human foot odor on worn socks (houses indoors).** (Source: Iman B. Hassaballa).

(Reynolds, Richmond, VA, USA) (Fig 3). In all cases, charcoal- filtered air was passed over the enclosed scented substrates at a flow rate of 350 ml/min on to two Super-Q adsorbents (30 mg, Analytical Research System, Gainesville, Florida, USA) for each replicate substrate. For each substrate, volatiles were collected for 12 h and replicated three–four times. Trapping from negative controls were also conducted: cleaned unused socks (human) or enclosed air or blank (dung and termite mound) employing the same conditions. Each adsorbent was eluted with 150 μL gas chromatography—mass spectrophotometer (GC-MS) grade dichloromethane (DCM) (Burdick and Jackson, Muskegon, Michigan, USA) and the eluent (300 μl/replicate substrate) stored at -80˚C until chemical analysis as described below.

## Analysis of volatiles

For identification of the constituent volatile organic compounds (VOCs) of the different substrates, an aliquot (1μl) of the dichloromethane (DCM) volatile extract of each sample (and controls) was injected into a gas chromatograph coupled mass spectrometer (GC-MS) in a splitless injection mode. The GC was equipped with an HP-5 column (30 m × 0.25 mm ID × 0.25 μm film thickness) with helium as the carrier gas at a flow rate of 1.2 ml/min. The oven temperature was held at 35˚C for 5 min, then programmed to increase at 10˚C/min to 280˚C and was maintained at this temperature for 10.5 min. The mass selective detector was maintained at ion source temperature of 230˚C and a quadrupole temperature of 180˚C. Electron impact (EI) mass spectra were obtained at the acceleration energy of 70 eV. Fragment ions was analysed over 40–550 $m/z$ mass range in the full scan mode. The filament delay time was set at 3.3 min. The volatile organic compounds were identified by comparing their mass spectra with library data (Adams2.L, Chemecol.L and NIST05a.L) search program (v. 2.0) and NIST Chemistry Webbook. Compounds present in controls were excluded from compositional profiles in each sample. Where available, the identities of VOCs were confirmed by co-

injection and comparison of mass spectral data with authentic standards. Retention indices (RI) were determined with reference to a homologous series of normal alkanes $C_8$-$C_{23}$ and calculated using the equation below as described by Van den Dool and Kratz [35] and comparison with published literature [36–38].

$$\mathbf{RIx = 100\ n_0 + 100(R_Tx - R_Tn_1)/(R_Tn_1 - R_Tn_0)}$$

With:

x = the name of the target compound

$n_1$ = n-alkane $C_{n1}H_{2n1+2}$ directly eluting before x

$n_0$ = n-alkane $C_{no}H_{2nO+2}$ directly eluting after x

$R_T$ = retention time

RI = retention index

## Chemicals

The chemicals used in GC-MS analysis including hexanal, heptanal, benzaldehyde, octanal, nonanal, decanal, 6-methyl-5-hepten-2-one, acetophenone, $\alpha$-phellandrene, $\alpha$-pinene, $p$-cymene, β-citronellene, camphene, sabinene, $\beta$-pinene, limonene, 1,8-cineole, ($Z$)-$\beta$-ocimene, ($E$)-$\beta$-ocimene, $\gamma$-terpinene, $\delta$-2- carene, ($Z$)-linalool oxide (furanoid), linalool, $\alpha$-copaene, $\alpha$-cedrene, ($E$)-caryophyllene, $\alpha$-humulene, ($Z$)- caryophyllene, skatole, heptanol, octanol, 1-octen-3-ol, nonanol, $p$-xylene, $o$-xylene, phenol, benzyl alcohol, $m$-cresol, $p$-cresol, indole, and standard n-alkanes solution were purchased from Sigma Aldrich (purity >97%).

## Data analysis

Daily catches were recorded for each sand fly species and sex per trap for each habitat type. Total sand fly abundance and specific-species abundance (inclusive of males and females) were analysed using generalized linear models (GLM) with a negative binomial error structure, with trapping period and habitat type specified as predictors. Abundance was compared for the leishmaniasis vectors *P. martini*, *P. duboscqi* and for four most abundant *Sergentomyia* species (*S. schwetzi*, *S. antennata*, *S. clydei* and *S. africana africana*). Total daily catches/trap by species (both sexes included) per habitat type were recorded from which measures of sand fly community structure Shannon diversity index (*H*, hereafter referred to as diversity) and species richness, were estimated and compared between the habitats and trapping period using generalized linear models. All analyses were conducted in R v. 3.6.3 [39] using the MASS package (for abundance data) and vegan package (diversity data) at 95% significance level. Best-fit models were selected based on model residuals. December 2018 and animal shed served as references for trapping period and habitat type, respectively. The mean of replicate peak areas of volatile compounds identified by GC-MS for dung, human socks and termite mound representing the different sand fly habitats, were visualised in a heatmap using the R software package "(gplots)" [40]. After checking for data normality using Shapiro-Wilk test ($P > 0.05$), we used analysis of variance (ANOVA) followed by Tukey's test to compare daily mean catches per trap for selected sand fly species between the habitats.

## Results

### Sand fly abundance and composition

A total of 6,931 sand flies (f = 4732, m = 2199) were collected comprising nine species in two genera across the two sampling periods (Table 1). In December 2018, 2952 sand flies were

**Table 1. Composition of sand flies caught across the three habitat types during the dry season in Rabai, Marigat sub-county, Kenya.**

| | | Animal shed | | | | House indoors | | | | Termite mound | | | |
|---|---|---|---|---|---|---|---|---|---|---|---|---|---|
| Trapping period | Sand fly species | M | F | Total | % | M | F | Total | % | M | F | Total | % |
| December 2018 | *P. duboscqi* | 11 | 18 | 29 | 4.89 | 0 | 0 | 0 | 0.00 | 8 | 14 | 22 | 1.91 |
| | *P. martini* | 29 | 17 | 46 | 7.76 | 7 | 7 | 14 | 1.16 | 70 | 57 | 127 | 11.05 |
| | *P. saevus* | 0 | 0 | 0 | 0.00 | 0 | 0 | 0 | 0.00 | 0 | 0 | 0 | 0.00 |
| | *S. adleri* | 4 | 1 | 5 | 0.84 | 0 | 0 | 0 | 0.00 | 0 | 2 | 2 | 0.17 |
| | *S. africana africana* | 23 | 7 | 30 | 5.06 | 61 | 79 | 140 | 11.57 | 14 | 13 | 27 | 2.35 |
| | *S. antennata* | 21 | 33 | 54 | 9.11 | 139 | 330 | 469 | 38.76 | 23 | 64 | 87 | 7.57 |
| | *S. clydei* | 109 | 115 | 224 | 37.77 | 31 | 24 | 55 | 4.55 | 44 | 98 | 142 | 12.36 |
| | *S. schwetzi* | 99 | 92 | 191 | 32.21 | 299 | 226 | 525 | 43.39 | 283 | 447 | 730 | 63.53 |
| | *S. squamipleuris* | 0 | 14 | 14 | 2.36 | 1 | 6 | 7 | 0.58 | 0 | 12 | 12 | 1.04 |
| | Total | | | 593 | | | | 1210 | | | | 1149 | |
| January 2020 | *P. duboscqi* | 9 | 12 | 21 | 1.96 | 1 | 2 | 3 | 0.29 | 4 | 18 | 22 | 1.17 |
| | *P. martini* | 19 | 15 | 34 | 3.18 | 9 | 13 | 22 | 2.13 | 28 | 30 | 58 | 3.09 |
| | *P. saevus* | 0 | 1 | 1 | 0.09 | 3 | 0 | 3 | 0.29 | 13 | 9 | 22 | 1.17 |
| | *S. adleri* | 0 | 1 | 1 | 0.09 | 0 | 1 | 1 | 0.10 | 0 | 2 | 2 | 0.11 |
| | *S. africana africana* | 12 | 51 | 63 | 5.89 | 8 | 70 | 78 | 7.54 | 6 | 35 | 41 | 2.19 |
| | *S. antennata* | 144 | 486 | 630 | 58.88 | 56 | 588 | 644 | 62.28 | 187 | 581 | 768 | 40.96 |
| | *S. clydei* | 11 | 26 | 37 | 3.46 | 15 | 33 | 48 | 4.64 | 18 | 84 | 102 | 5.44 |
| | *S. schwetzi* | 82 | 159 | 241 | 22.52 | 75 | 109 | 184 | 17.79 | 210 | 575 | 785 | 41.87 |
| | *S. squamipleuris* | 6 | 36 | 42 | 3.93 | 5 | 46 | 51 | 4.93 | 2 | 73 | 75 | 4.00 |
| | Total | | | 1070 | | | | 1034 | | | | 1875 | |

M = males; F = females

collected with female catches significantly higher than males (f = 1676, m = 1276; $\chi^2$ = 53.93, df = 1, p < 0.0001). The 3,979 sand flies collected in January 2020 comprised 923 males and 3056 females with significant difference in catches between the sexes ($\chi^2$ = 1142.4, df = 1, p < 0.0001). Sand flies in the *Phlebotomus* genus were represented by three species: *P. duboscqi*, *P. martini* and *P. saevus*. Six species encountered in the genus *Sergentomyia* included *S. adleri*, *S. africana africana*, *S. antennata*, *S. clydei*, *S. schwetzi* and *S. squamipleuris* (Table 1).

Overall, sand fly catches were predominantly *Sergentomyia* species, notably *S. schwetzi* (38.3%), *S. antennata* (38.3%) and *S. clydei* (8.8%). *P. martini* comprised 4.3% of the total captures while *P. duboscqi* and *P. saevus* accounted for 1.4% and 0.4%, respectively (Table 1).

### Habitat specific abundance patterns

Of the leishmaniasis vectors, *P. martini* was the most abundant *Phlebotomus* species collected from all the habitats during the trapping period. *P. duboscqi* was not encountered in the collections from houses indoors in December 2018. *P. saevus* implicated recently as CL vector [7] was encountered in the January 2020 collection across the habitats although predominantly from termite mounds (Table 1). Regardless of the trapping session, collections from each habitat type was dominated by *Sergentomyia* species. In the December 2018 collection from animal shed, *S. clydei* and *S. schwetzi* were the most abundant species. Predominant species were *S. schwetzi* followed by *S. antennata* in houses indoors with the former species recorded in highest numbers in termite mound (Table 1 and Fig 4). In the January 2020 collection, *S. antennata* was the most abundant species collected from the animal shed followed by *S. schwetzi*, with both species dominating catches obtained from houses indoors. The most abundant species collected from termite mound was *S. schwetzi* and *S. antennata* (Table 1 and Fig 4).

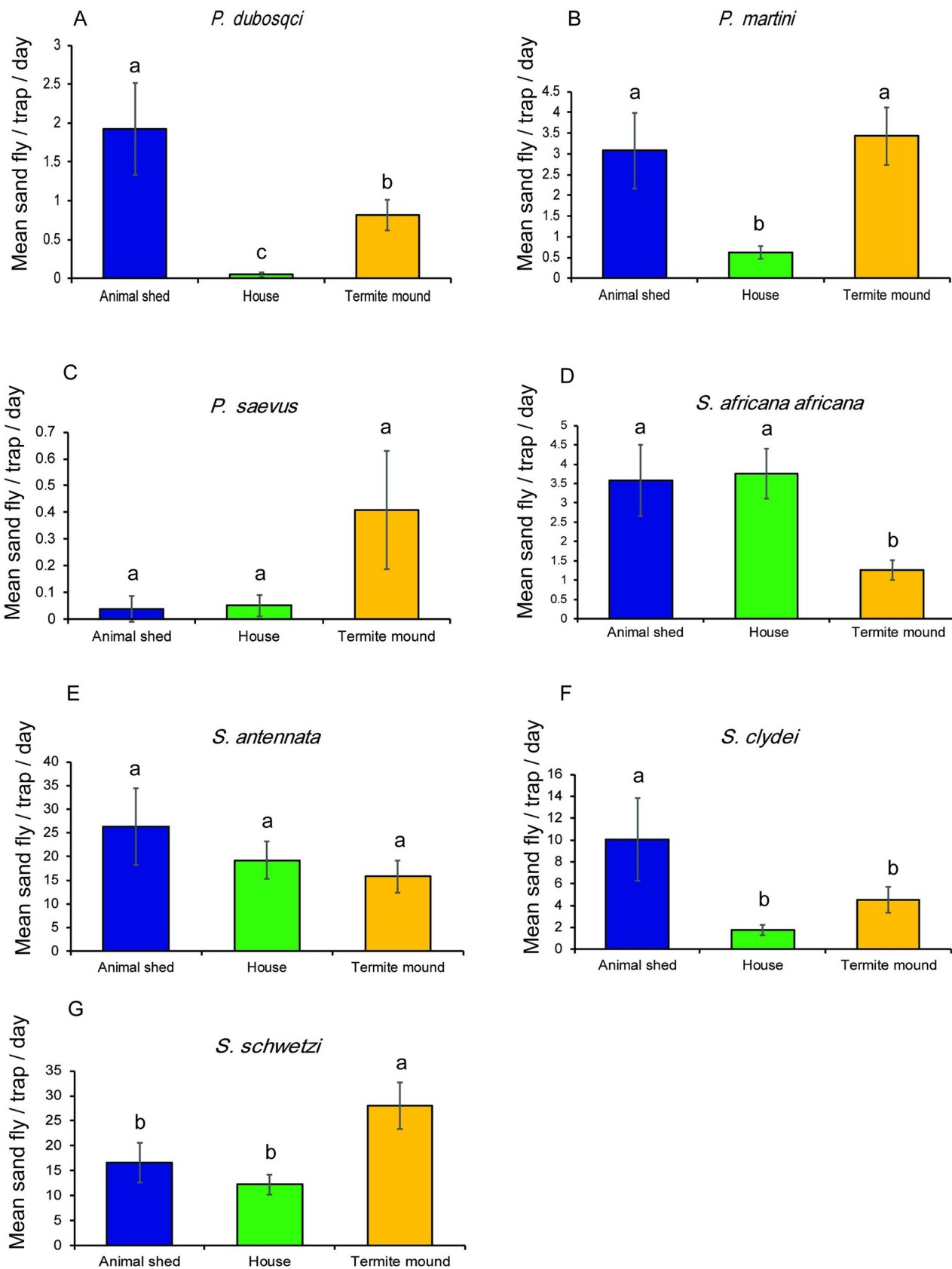

**Fig 4. Mean numbers (± SE) of sand fly species collected in CDC light traps/day/night from three different sampling habitats in two trapping sessions in Rabai village, Marigat sub-county.** Means followed by different letters are significantly different at α = 0.05 according to Tukey's test of ANOVA.

Analysis of species-specific habitat patterns indicated that abundance of *P. martini* was highest for termite mounds and then animal shed, which was significantly different from catches in houses indoors (Table 2). There was a 5-fold chance of this species being encountered in termite mound and animal shed than in houses indoor (Fig 4). Significantly higher catches of *P. duboscqi* was found in animal sheds than termite mounds and houses indoors. The likelihood of *P. duboscqi* being captured in animal shed was 38- and 2-times higher than in houses indoors and termite mound, respectively. The mean number of catches for each species by habitat is presented in Fig 4.

Abundance of *S. antennata* did not vary by habitat although higher mean catches were obtained from the animal shed than in houses indoors and termite mound. No significant difference in catches was observed between the habitats for *S. antennata*, however, mean catch recorded for *S. schwetzi* was higher from the termite mounds, than from the animal shed and houses indoors. On the other hand, the catch for *S. clydei* was highest from animal shed, which significantly varied from that recorded for houses indoors but not for termite mound in both species. Among the species, variation in abundance based on the trapping period was evident for *P. saevus* and *S. antennata* (Table 2).

**Table 2. Sand fly abundance and diversity trends sampled during the dry season in three habitat types in Rabai, Baringo County, Kenya.** Models used were GLMs with negative binomial error structure. Animal shed served a reference category for habitat and January 2020 for trapping period.

| | Total abundance (df = 5,134) | | | *P. duboscqi* abundance (df = 5,134) | | |
|---|---|---|---|---|---|---|
| | Estimate ± SE | Z value | P-value | Estimate ± SE | Z value | P- value |
| House | -0.421 ± 0.251 | -1.676 | 0.094 | -3.621 ± 0.686 | -5.277 | < 0.001 *** |
| Termite mound | -0.121 ± 0.254 | -0.476 | 0.634 | -0.860± 0.402 | -2.14 | 0.032 * |
| Period: January 2020 | 0.447 ± 0.183 | 2.451 | 0.014 * | 0.152 ± 0.373 | 0.407 | 0.684 |
| | *P. martini* abundance (df = 5,137) | | | *P. saevus* abundance (df = 4,135) | | |
| | Estimate ± SE | Z value | P- value | Estimate ± SE | Z value | P- value |
| House | -1.595 ± 0.381 | -4.093 | <0.001 *** | 0.296± 1.444 | 0.205 | 0.837 |
| Termite mound | 0.095 ± 0.364 | 0.26 | 0.795 | 2.360 ± 1.347 | 1.752 | 0.079 |
| Period: January 2020 | -0.100 ± 0.283 | -0.354 | 0.723 | - | - | - |
| | *S. africana africana* abundance (df = 5,134) | | | *S. antennata* abundance (df = 5,134) | | |
| | Estimate ± SE | Z value | P- value | Estimate ± SE | Z value | P- value |
| House | 0.116 ± 0.315 | 0.368 | 0.713 | 0.426 ± 0.332 | 1.284 | 0.199 |
| Termite mound | -1.017±0.334 | -3.046 | 0.002 ** | -0.411 ±0.338 | -1.241 | 0.214 |
| Period: January 2020 | 0.211 ± 0.239 | 0.919 | 0.358 | 1.696 ± 0.242 | 7.001 | < 0.001*** |
| | *S. clydei* abundance (df = 5,134) | | | *S. schwetzi* abundance (df = 5,134) | | |
| | Estimate ±SE | Z value | P- value | Estimate ±SE | Z value | P- value |
| House | -1.636±0.464 | -3.529 | 0.0004 *** | -0.347±0.296 | -1.17 | 0.242 |
| Termite mound | -0.713±0.463 | -1.542 | 0.123 | 0.518 ± 0.298 | 1.736 | 0.083 |
| Period: January 2020 | -0.308±0.341 | -0.903 | 0.367 | -0.135±0.215 | -0.626 | 0.531 |

*$P < 0.05$

**$P < 0.01$

***$P < 0.001$

**Table 3. Sand fly diversity trends during the dry season in three habitat types in Rabai, Baringo County, Kenya.** Models used were GLMs. Animal shed served a reference category for habitat and January 2020 for trapping period.

| | Shannon diversity index (df = 5,134) | | | Species richness (df = 5,134) | | |
|---|---|---|---|---|---|---|
| | Estimate ± SE | t value | P- value | Estimate ±SE | t value | P- value |
| House | -0.416 ± 0.091 | -4.579 | <0.001*** | -1.937±0.378 | -5.118 | <0.001*** |
| Termite mound | -0.234 ± 0.092 | -2.542 | 0.012* | -0.639±0.383 | -1.669 | 0.097. |
| Period: January 2020 | 0.019 ± 0.066 | 0.288 | 0.774 | 0.900 ± 0.275 | 3.273 | < 0.001*** |

*$P < 0.05$

***$P < 0.001$

## Sand fly diversity patterns

There was variation in species richness by habitat and sampling period. Animal shed had the highest species richness, which significantly differed from house but not termite mound. Sand fly diversity was highly influenced by habitat and trapping period. Like richness, diversity was highest for animal shed ($H = 1.2$), which varied significantly from values observed for the termite mound ($H = 1.0$) or house ($H = 0.8$) (Table 3 and Fig 5).

## Analysis of volatiles

Analysis of volatiles collected from the three substrates representative of animal shed (fresh cow dung), termite mound and houses indoors (human foot odors) by GC-MS detected 47 VOCs. Of these volatiles, 26 were detected from human worn socks, 35 from cow dung and 16 from termite mound odors. The compositional profile of the VOCs detected across the three odor substrates are represented in a heatmap (Fig 6). The volatiles generally belonged to seven functional groups: aldehyde, alcohol, benzenoid, ketone, monoterpene, nitrogenous comp and sesquiterpene (Table 4). Eight of these VOCs including α-pinene, 1-octen-3-ol, 6-methyl-5-hepten-2-one (sulcatone), limonene, benzyl alcohol, *m*-cresol, *p*-cresol and decanal (Fig 6 and Table 4) were common to the three volatile sources.

## Discussion

Knowledge of ecologic factors influencing vector distribution can provide insights into disease epidemiology and avenues for control. This study investigated the distribution of sand flies in selected habitats in an endemic site for leishmaniasis in Kenya. Additionally, the possible olfactory determinants of their selection of these habitats was investigated. Our results indicate that habitat type influences the diversity and abundance of sand fly species. Interestingly, the overall abundance did not vary between these habitats; however, species-specific differences in abundance were evident. Previous research showed that *P. martini* is associated with termite mounds [22,23]. Despite habitat selection having been suggested for sand flies in Kenya, to the best of our knowledge, this is the first report comparing sand fly diversity and richness profiles across selected habitats in Kenya. Intriguingly, we found that both measures of community structure (diversity and richness) were highest in animal shed, followed by termite mound and lowest in houses indoors. Our findings further lend support for habitat choice among sand fly species which can potentially be exploited in the control of leishmaniasis or perhaps other diseases they transmit to break transmission to humans.

Notably, we found that overall, more females were captured than males across habitats throughout the study period. Our results are consistent with previous findings of sand fly collections using light traps [7,22]. The pattern can be explained considering our trapping times,

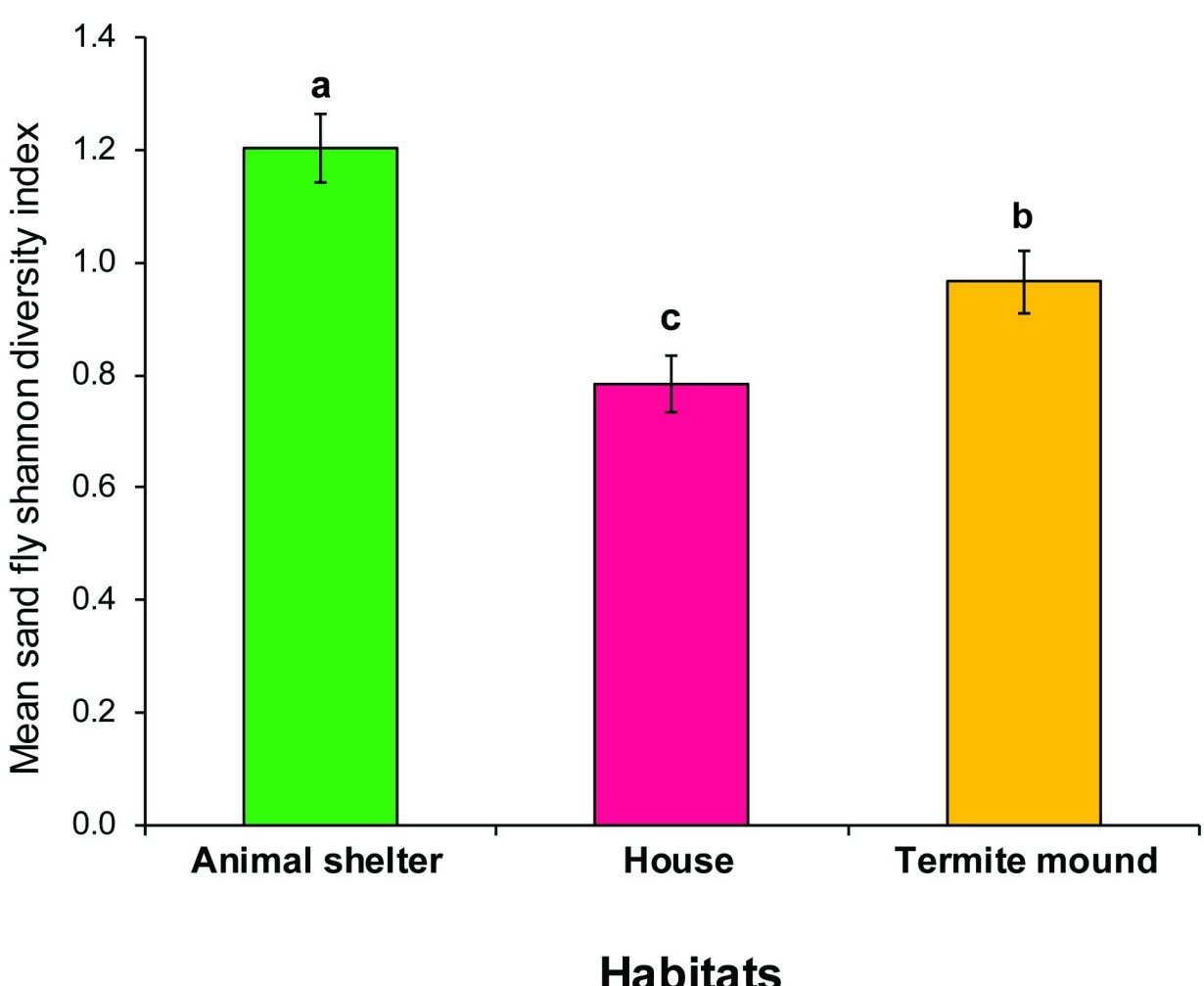

**Fig 5. Mean Shannon diversity index for Phlebotomine sand flies during the dry season in Rabai, Marigat sub-County, Baringo County, Kenya.** Sand flies were surveyed using CDC light traps. Means followed by different letters are significantly different at $\alpha$ = 0.05 according to Tukey's test of ANOVA.

which occurred between 18:00 h—06:00 h. These are times when sand flies, predominantly females actively search for hosts for a blood meal, required for maturation of their eggs [41]. Although sand flies were not trapped during the wet season, our results suggest that abundant phlebotomine sand flies species occur in this ecology during the trapping periods in the dry season as previously noted by other investigators in Kenya [25,42], Sudan [17] and Ethiopia [43]. Regarding the examined vectors of VL and CL, their presence across the habitats potentially indicates the ease of movement between these habitats to feed on human and domestic animals or reservoir hosts in resting sites as previously noted [42]. This would have epidemiological implications and hence explain the maintenance of the transmission of leishmaniasis between animals and man. A similar analogy pertains to sand flies in the *Sergentomyia* genus which have recently been implicated in the transmission of a novel virus of potential zoonotic importance and infecting humans in this ecology [44]. Perhaps, a variation in habitat microclimatic factors such as temperature, moisture and organic matter known to influence sand fly abundance [45] may have contributed to the differential abundance trends observed between the two trapping periods.

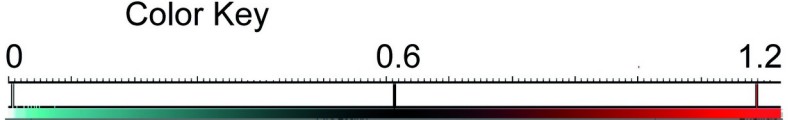

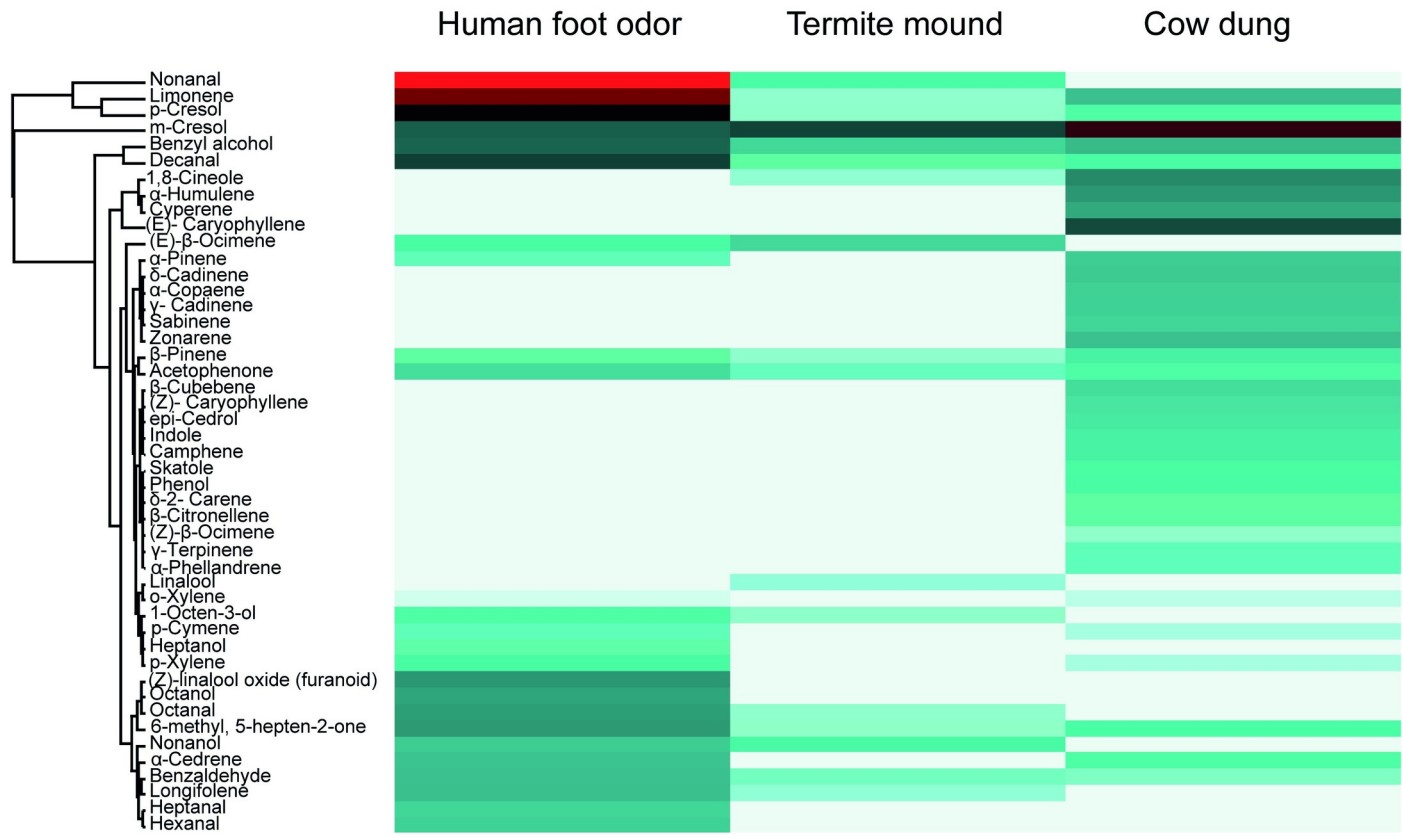

**Fig 6. Heatmap depicting the volatile organic compounds (VOCs) identified from representative substrates of habitat types: human foot odor (houses indoors), termite mound and fresh cow dung (animal shed).**

Suitable habitats are important in disease epidemiology as they influence the life history parameters of vectors e.g. development and population growth rates, resting, and breeding. Resting sites provide a microclimate to extend vector survival thereby contributing to their ability to transmit pathogens. The contrasting patterns of habitat selection exhibited among the two *Phlebotomus* vector species, that is, *P. martini* and *P. duboscqi* and even between the *Sergentomyia* species examined are of ecological significance. Several propositions could be advanced for the difference in habitat preferences. First, the fundamental differences in the biology between the species likely reflects utilization of the habitats. Some sand fly vectors are dependent on their blood-feeding host as a source of habitat. For instance, *P. duboscqi* and *P. papatasi* which are both vectors of *Leishmania major* preferentially rest and breed within burrows of their rodent blood meal hosts and reservoir of the parasite [23,46,47]. *P. duboscqi* mainly bites humans when they occur in areas with many animal burrows [23,25]. Blood meal analysis previously carried out in the study site showed that *P. martini* fed more commonly on cow but also humans and dogs, similarly to *S. schwetzi* [48]. *S. antennata* which primarily feeds on reptiles has been shown to also bite humans [48]. While many *Sergentomyia* species

**Table 4. Summary of identified volatile organic compounds (VOCs) from three Phelobtomine sand fly habitats in Rabai, Marigat sub-county, Baringo County, Kenya.**

| RT (min) | Compound name | Cow dung | Termite mound | Human foot odor on worn socks | Functional group | RI[a] | RI[b]$_L$ |
|---|---|---|---|---|---|---|---|
| 6.49 | Hexanal | - | - | + | Aldehyde | 803 | 801 |
| 8.32 | p-Xylene | + | - | + | Benzenoid | 838 | 884 |
| 8.89 | o-Xylene | + | - | + | Benzenoid | 884 | 888 |
| 9.12 | Heptanal | - | - | + | Aldehyde | 903 | 907 |
| 9.69 | α-Phellandrene | + | - | - | Monoterpene | 949 | 985 |
| 9.82 | α-Pinene* | + | + | + | Monoterpene | 960 | 932 |
| 10.09 | β-Citronellene | + | - | - | Monoterpene | 982 | 938 |
| 10.14 | Camphene | + | - | - | Monoterpene | 985 | 953 |
| 10.43 | Benzaldehyde | + | - | + | Aldehyde | 1009 | 965 |
| 10.65 | Heptanol | - | - | + | Alcohol | 1027 | 974 |
| 10.67 | Sabinene | + | - | - | Monoterpene | 1029 | 1017 |
| 10.73 | β-Pinene | + | - | + | Monoterpene | 1033 | 1008 |
| 10.81 | 1-Octen-3-ol* | + | + | + | Alcohol | 1040 | 1456 |
| 10.88 | Phenol | + | - | - | Benzenoid | 1046 | 1050 |
| 10.97 | 6-Methyl-5-hepten-2-one* | + | + | + | Ketone | 1053 | 987 |
| 11.28 | Octanal | - | + | + | Aldehyde | 1007 | 1009 |
| 11.67 | p-Cymene | - | + | + | Monoterpene | 1041 | 1020 |
| 11.75 | Limonene* | + | + | + | Monoterpene | 1051 | 1031 |
| 11.80 | 1,8-Cineole | + | + | - | Monoterpene | 1057 | 1031 |
| 11.86 | Benzyl alcohol* | + | + | + | Benzenoid | 1062 | 1017 |
| 11.92 | (Z)-β-Ocimene | + | - | - | Monoterpene | 1021 | 1044 |
| 12.10 | (E)-β-Ocimene | - | + | + | Monoterpene | 1038 | 1054 |
| 12.30 | γ-Terpinene | + | - | - | Monoterpene | 1053 | 1076 |
| 12.46 | Acetophenone | + | + | + | Ketone | 1057 | 1561 |
| 12.51 | Octanol | - | - | + | Alcohol | 1061 | 1098 |
| 12.56 | m-Cresol* | + | + | + | Benzenoid | 1067 | 1077 |
| 12.63 | p-Cresol* | + | + | + | Benzenoid | 1085 | 1001 |
| 12.83 | δ-2-Carene | + | - | - | Monoterpene | 1085 | 1068 |
| 12.83 | (Z)-linalool oxide (furanoid) | - | - | + | Monoterpene | 1100 | 1095 |
| 13.00 | Linalool | - | + | - | Monoterpene | 1107 | 1087 |
| 13.08 | Nonanal | - | + | + | Aldehyde | 1133 | 1014 |
| 14.16 | Nonanol | - | - | + | Alcohol | 1161 | 1186 |
| 14.71 | Decanal* | + | + | + | Aldehyde | 1211 | 1203 |
| 16.02 | Indole | + | - | - | Benzenoid | 1298 | 1298 |
| 17.20 | α-Copaene | + | - | - | Sesquiterpene | 1383 | 1378 |
| 17.30 | Skatole | + | - | - | Nitrogenous comp. | 1392 | 1381 |
| 17.57 | Cyperene | + | - | - | Sesquiterpene | 1416 | 1401 |
| 17.64 | (Z)- Caryophyllene | + | - | - | Sesquiterpene | 1423 | 1421 |
| 17.65 | Longifolene | - | + | + | Sesquiterpene | 1425 | 1406 |
| 17.73 | α- Cedrene | + | - | + | Sesquiterpene | 1432 | 1413 |
| 17.81 | (E)- Caryophyllene | + | - | - | Sesquiterpene | 1439 | 1417 |
| 17.92 | β-Cubebene | + | - | - | Sesquiterpene | 1449 | 1460 |
| 18.25 | α- Humulene | + | - | - | Sesquiterpene | 1462 | 1454 |
| 18.76 | Zonarene | + | - | - | sesquiterpene | 1510 | 1521 |
| 18.97 | γ- Cadinene | + | - | - | Sesquiterpene | 1530 | 1513 |
| 19.07 | δ-Cadinene | + | - | + | Sesquiterpene | 1539 | 1523 |

(Continued)

**Table 4.** (Continued)

| RT (min) | Compound name | Cow dung | Termite mound | Human foot odor on worn socks | Functional group | RI$^a$ | RI$^b_L$ |
|---|---|---|---|---|---|---|---|
| 20.10 | epi-Cedrol | + | - | - | Sesquiterpene | 1624 | 1611 |

(RT) = retention times.

RI$^a$ = Retention index relative to C8-C23 n- alkanes of a HP-5 MS column.

RI$^b_L$ = Retention index obtained from literature: [36–38].

(+) = present compound and (-) = absent compound.

*Compounds detected in all three different substrates.

are regarded as catholic in feeding habits, recent findings suggest the readiness of some species to expand their host range to include feeding on humans [44]. Thus, the overall high sand fly abundance and diversity in animal shed may reflect presence of livestock hosts as a potential source of blood meal, with ease of access compared to human habitations or termite mounds. The differential presence of sand flies across the habitats could be an indication about potential differences in host range of the species (narrow vs wide range). Second, the represented substrates of the different habitat types release volatiles that may serve as olfactory signals to these insects to which they may respond differently. Variation in response to volatile signals among different species of hematophagous insects like mosquitoes is known in literature [28]. For instance, sand flies may associate cues from mounds/animal sheds for suitability as breeding sites for egg laying or development of immatures [49]. Organic matter associated with some of these sites used for breeding [45], may serve as sources of volatiles to ascertain suitability of the sites. Volatile cues may be used as habitat location cues [50]. Sand flies must also host-seek and bite humans to transmit pathogens, a process widely known to be largely influenced by olfactory cues.

We explored the latter proposition by analysing the VOCs from substrates associated with these habitats. Interestingly, we detected a total of 47 VOCs although there was more representation from human worn socks (26/47) and cow dung (35/47) than from termite mounds (16/47). However, of particular interest are VOCs common across the habitats including 1-octen-3-ol, 6-methyl-5-hepten-2-one (sulcatone), α-pinene, limonene, benzyl alcohol, *m*-cresol, *p*-cresol and decanal (Table 4). Laboratory and field studies with some of these compounds including 1-octen-3-ol and 6-methyl-5-hepten-2-one have shown increased responses from different sand fly species, which confirms their roles as attractants in sand fly chemical ecology. For instance, in laboratory assays, Magalhães- Junior et al [51] demonstrated the attractiveness of decanal and nonanal to the sand fly species, *Lutzomyia longipalpis*. Octenol is a known attractant for a wide range of blood feeding insects including sand flies [52] and has been reported in the emanations from diverse sources e.g., cattle [53], human breath [54], human skin [55] and plants [56]. For the alcohols, hexanol and octanol have been found to elicit attractive responses in sand flies in laboratory and field settings [52,57]. A recent finding implicated the human-specific cue, sulcatone [28], in the attractive response of the New World sand fly *Lutzomyia intermedia* in laboratory assays [27]. Further, a subset of these compounds has been reported to play a role in the olfactory and behavioural ecology of other blood feeding insects like mosquitoes (e.g., 6-methyl-5-hepten-2-one, heptanal, decanal, nonanal, α-pinene, indole) [26], and tsetse flies (e.g., 6-methyl-5-hepten-2-one, heptanal, octanal, nonanal and decanal) [30]. For instance, Baleba et al [29] found that limonene and *m*-cresol were the most important VOCs of cow dung eliciting attraction in the stable fly *Stomoxys calcitrans* for egg laying. The already proven role of some of these compounds as attractants lends support to proposition that they could be involved in habitat selection among sand flies. A review on

sand fly biology noted that seasonality in the types of resting sites used by these flies is a common occurrence likely influenced by the amount of moisture [58]. However, this is likely to be insufficient to represent an independent mechanism of habitat selection. Noteworthy, whereas terpenes were found to be more represented in volatiles of cow dung, aldehydes dominated human volatiles. The different classes of volatiles may interact with each other in specific ways or ratios important in modulating attraction to a given substrate. Nonetheless, the current study provides empirical evidence of possible involvement of volatiles in habitat selection.

While the hypothesis about role of semiochemicals in habitat selection among sand flies is tentatively indicative by our data, additional studies are needed, including using different volatile sampling techniques, to establish a clear association of sand fly catches with specific volatile profiles from the examined substrates. Such considerations should capture the contribution of background habitat volatiles. Another methodological challenge included the difficulty to access habitat types without sand flies for comparative analysis of volatile profiles. Thus, we limited our assessments to the sites we confirmed had sand flies based on trap collections. Other sand fly habitats not captured in this study should be the focus of future research.

In conclusion, the present findings show differential abundance and diversity trends among *Phlebotomus* and *Sergentomyia* sand fly species, vectors of leishmaniasis and arboviruses in three selected habitats during the dry season. This knowledge on habitat choice can potentially be exploited in adult sand fly control or management to break disease transmission, especially during emergency outbreak situations. We note that these habitats or associated substrates emit important olfactory cues which could influence the observed choices, as some have proven roles as attractants for sand flies and other blood feeding insects. Evaluating these compounds for their behavioral impact will inform the basis for their development as lures to improve sand fly monitoring and even control. Overall, our findings offer insights into habitat selection among sand fly species and add to the existing body of literature about aspects of sand fly bioecology and epidemiology of the diseases they are associated with.

## Acknowledgments

We are thankful to Mr Onesmas Wanyama and Pascal Ayelo, Behavioral and Chemical Ecology Unit, *icipe*, Nairobi, for technical support. We are grateful to Jackson Kimani and Emily Kimathi, GIS support unit, *icipe*, for creating the map of the study site. We appreciate the support of the chief and community members of Rabai, and local administration of Marigat sub-county especially Mr Mark Rotich for support and cooperation throughout the study.

## Author Contributions

**Conceptualization:** Baldwyn Torto, David P. Tchouassi.

**Data curation:** Iman B. Hassaballa.

**Formal analysis:** Iman B. Hassaballa, David P. Tchouassi.

**Funding acquisition:** Baldwyn Torto, David P. Tchouassi.

**Investigation:** Iman B. Hassaballa.

**Methodology:** Iman B. Hassaballa, Baldwyn Torto, David P. Tchouassi.

**Resources:** Baldwyn Torto, David P. Tchouassi.

**Supervision:** Baldwyn Torto, Catherine L. Sole, David P. Tchouassi.

**Validation:** Baldwyn Torto, Catherine L. Sole, David P. Tchouassi.

**Writing – original draft:** Iman B. Hassaballa, Baldwyn Torto, David P. Tchouassi.

**Writing – review & editing:** Iman B. Hassaballa, Baldwyn Torto, Catherine L. Sole, David P. Tchouassi.

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
