## [Decision Letter · Decision Letter 0]

16 Sep 2020

Dear Dr. Tchouassi,

Thank you very much for submitting your manuscript "Exploring the influence of different habitats and their volatile chemistry in modulating sand fly population structure in a leishmaniasis endemic foci, Kenya" for consideration at PLOS Neglected Tropical Diseases. As with all papers reviewed by the journal, your manuscript was reviewed by members of the editorial board and by several independent reviewers. In light of the reviews (below this email), we would like to invite the resubmission of a significantly-revised version that takes into account the reviewers' comments. 

We cannot make any decision about publication until we have seen the revised manuscript and your response to the reviewers' comments. Your revised manuscript is also likely to be sent to reviewers for further evaluation.

Sincerely,

Geraldine Marie Foster, PhD

Guest Editor

Helen Price

Deputy Editor

Reviewer's Responses to Questions

**Key Review Criteria Required for Acceptance?**

**Methods**

-Are the objectives of the study clearly articulated with a clear testable hypothesis stated?

-Is the study design appropriate to address the stated objectives?

-Is the population clearly described and appropriate for the hypothesis being tested?

-Is the sample size sufficient to ensure adequate power to address the hypothesis being tested?

-Were correct statistical analysis used to support conclusions?

-Are there concerns about ethical or regulatory requirements being met?

Reviewer #1: Because representative substrates were used to characterize the VOCs from the animal shed and house, it is difficult to assess the relationship between the sand fly epidemiologic data and the habitat-specific VOC profiles without more information:

1. For the animal dung used as substrate for the animal shed VOCs: was the dung collected from the same animal sheds and during the same time periods where the sand fly collections took place? If so, please clarify this in the text. If not, please explain why this did not occur. If you have information about the animal species from which the dung originated, please also include this information. 

2. For the human-worn socks used as a substrate for VOC detection: were these socks worn by the same humans living in the houses where sand fly collection took place, and during the same time periods? If so, please clarify this in the text. If not, please explain why this did not occur.

3. The data does not convey the inter-individual variability in VOC profiles (both in humans and from the animal dung). Please show data that makes explicit which VOCs were specific to a minority of samples from one habitat type, versus VOCs which were present in the majority or all the samples from one habitat. This is important to assess how representative or generalizable the VOC profile is for a given habitat. 

4. Why were VOCs not directly captured from the air in the animal shed and house habitats? The authors showed they could successfully capture odorants directly from the termite mounds. Direct capture of VOCs from the environment would help mitigate concerns about the generalizability of the VOC data gathered from proxy substrates (as raised in points 1 through 3 above). If there are intractable technical obstacles to such an experiment, please discuss these considerations in the text.

Reviewer #2: -Are the objectives of the study clearly articulated with a clear testable hypothesis stated?

This can be improved see my attached comments

-Is the study design appropriate to address the stated objectives?

There are some problems in this area, see attached comments.

-Is the population clearly described and appropriate for the hypothesis being tested?

Not applicable.

-Is the sample size sufficient to ensure adequate power to address the hypothesis being tested?

Not entirely clear – see comments.

-Were correct statistical analysis used to support conclusions?

See comments

-Are there concerns about ethical or regulatory requirements being met?

No

**Results**

-Does the analysis presented match the analysis plan?

-Are the results clearly and completely presented?

-Are the figures (Tables, Images) of sufficient quality for clarity?

Reviewer #1: - Please discuss in the text possible reasons for why total sand fly abundance was higher in the January 2020 collection than December 2018 (Table 2). 

- Lines 304-305: “Among the species, seasonality based on trapping period only influenced catches of P. saevus and S. antennata (Table 2).” I don’t think “seasonality” is the appropriate term here, since January and December are very close to each other in the calendar. Please clarify this language.

Reviewer #2: -Does the analysis presented match the analysis plan?

Not applicable

-Are the results clearly and completely presented?

Nearly

-Are the figures (Tables, Images) of sufficient quality for clarity?

yes

**Conclusions**

-Are the conclusions supported by the data presented?

-Are the limitations of analysis clearly described?

-Do the authors discuss how these data can be helpful to advance our understanding of the topic under study?

-Is public health relevance addressed?

Reviewer #1: - Lines 462-463: The authors state that, “the current study provides empirical evidence of possible involvement of volatiles in habitat selection,” however the data provided do not fully support this conclusion, as currently phrased. This argument is further weakened by the use of proxy substrates for the habitat VOCs, as discussed above. To support this claim, the authors need to show an association between the habitat-specific abundance of different sand fly species and habitat-specific VOCs or VOC profiles. In the absence of such data, the authors can state that this is a limitation of the study, and that determining an association between the sand fly species abundance and VOC profiles reported in this study remains to be performed.

Reviewer #2: please see comments

**Editorial and Data Presentation Modifications?**

Reviewer #1: Other than the modifications discussed above, no other modifications needed.

Reviewer #2: in some areas explanations and grammar can be improved. many of these are minor issues but would help improve the overall presentation of the paper.

**Summary and General Comments**

Reviewer #1: REVIEW SUMMARY

Hassaballa et al. report the results of a cross-sectional survey of sand fly species abundance and diversity in Rabai village of Baringo County, Kenya, an endemic area for both visceral and cutaneous leishmaniasis. The study focuses on comparing sand fly species and diversity in three different kinds of habitats in the village – animal sheds, termite mounds, and houses. The authors hypothesize that sand fly habitat selection is due in part to different profiles of volatile organic compounds (VOCs) that act as odorants in the different habitats. The authors use LC-MS to catalogue VOCs directly from the termite mounds, and from proxy substrates of the animal sheds and houses, namely animal dung and human-worn socks, respectively. Overall, the study presents useful data for sand fly epidemiology and leishmania vector control efforts in East Africa, however minor revisions are necessary before I can recommend the manuscript for publication, as detailed in my critiques in the above sections.

Reviewer #2: This is an interesting study combining field observations with laboratory analysis. The study addresses important considerations in relation to east Africa vectors of Leishmaniasis. This type of work is severely lacking and therefore this work address major gaps in our knowledge.

My main concern is that the background and methodology of the work needs further explanation and clarification. The results also require clarification and the conclusions and Discussion should clearly reflect the limitations of the study.

However I do not want the authors to feel that this reviewer does not appreciate the efforts that they have gone to. Field work in this area is difficult and any contribution is valuable. With some small effort they can make this a very good and useful paper.

specific comments 

L2 Sand fly

L12 Missing author affiliations

L35 “guilds" = Communities?

L41-43 What does this mean – a greater number of species present in animal sheds?

L44-46 This is not clear Human houses = socks? Animal sheds =dung? Termite mounds =????

L51-53 It is not clear how this conclusion is arrived at. It would be acceptable (and more accurate) is you replace “show” with “suggest” in this sentence.

L69 understanding the ecology of the vector is essential.

L78 Did you formally try to associate species abundance or numbers with volatile abundance or diversity.

I mean in houses or animal shelters the increased sand fly species diversity and abundance might be related to the greater quantity of compounds present in those collecting sites compared to the termite mounds. Also those sites are likely to present a greater number of other sand fly relevant resources compared to the termite mounds. Why choose termite mounds?

L106 It is not clear what you mean here. Is vector control used against leishmania vectors in Kenya or not? I think you need to examine the Kenyan Ministry of Health policy in this area and report it here.

L108 Give an example and a reference

L108 Give an example and a reference

L110 “ In response to leishmaniasis outbreaks, spraying of houses with insecticides is common … “ This is contradicting the statement in L106 so this whole section needs to be clarified.

L111 “Thus, integrated vector management strategies for prevention and control of the sand fly vectors of the disease remain limited” Vague and not consistent with previous statements.

L113 This is not clear –reduction of human transmission is the objective of vector control.

L113 "… which can be achieved through improved understanding of sand fly behavioral adaptations” I’m not sure that t this is a logical consequence of the first part of the sentence.

L114 Are you talking about the situation in Kenya or East Africa or more generally?

L115 “…insecticides or other biocontrol agents …” I think there is room (and need) to be specific – give examples. 

L119 Which ecological trait?

L120-122 The authors use 2 citations (both are cited incorrectly) to support this point, unfortunately neither of the citations support the statement. Perhaps they suggested this option in the discussion? In any case the authors need to very carefully check the citations and make sure that actually support the points being made. The authors need to distinguish between actual (i.e. real control methods performed on a routine basis by recognised agencies and proposed or experimental control methodologies.

L113-129 The section on control is very weak. It needs to be rewritten there are plenty of good examples from India/Bangladesh, Iran and South America particularly Brazil.

L141 The sentence structure is not clear – are you talking about 2 foci? Or one focus of leishmaniasis?

L141 “..guilds..” It would be helpful to define what this term means in the context of this work.

L142-143 What is the basis for this hypothesis? You have not linked sand fly distribution with different odour previously in the introduction therefore you need to have a section before you define the objective of the study that sets out the logical argument for conducting this work. 

L146-148 This is an association study and therefore you cannot truly say that distribution is related to odour. This can only be done by a manipulative experiment. Please recognise this in the Discussion.

L157 Balanites spp.

L170 use “handheld GPS device” as it is better grammatically.

L179 How were the sites selected? This is an important consideration and you need to elaborate the point – also how many of each?

L180-181 Poor expression

L181 How many traps in each trapping site?

L181-182 Do you mean inside houses? – you should stick with your previous definition L147 of trapping sites.

L187-189 Why not store them in ethanol?

L197 Is there something missing here? does gum chloral clear the preparation if so how are they fixed on the glass side?

L206. This is a substantially different way of collecting the volatiles. And one immediate consequence that I can think of is that you would be likely to collect less volatiles from the termite mounds compared to the other 2 odour sources. I wonder why you did not use a standard so that you can directly compare the entrainments. Please add this point to your Discussion as well as commentary on the weaknesses in the methodology.

L 207 How many?

L209 details of solvent and oven temperature

L229 delete “to”

L235 replace “was” with were obtained

L239 Confirmation would require comparison of retention time on 2 columns of differing polarity. You need to moderate your claim of “confirmed structures.

L254 for “treatment” do you mean location?

L260 this is a Discussion point or could be included in the Introduction as a justification for collecting Sergentomyia.

L267 this is not clear please explain and rephrase

L278 I am assuming verbal? – please confirm.

L289 P.

L302 this is an interesting word which implies that is was caught elsewhere in your study. I think it’s worth giving the details.

L303 “…trapping season …” I don’t think you can describe these collection time points as different “trapping seasons”. One collection time point is December the other in January – these are in the same season even though separate in time. It would have been interesting to collect in a different season. You actually refer to the season “dry season” later in the manuscript.

L314 “…than in houses indoor …” replace with, inside houses and in other places where you use this style.

L380 Table 4 this table doesn’t tell me how many compounds that were present in each habitat were not identified. Some of the “identifications” are really quite suspect. I’m not really sure why you need to try and identify all the compounds. You can note absence/presence, quantities and total numbers of compounds and base your comparison of these sites on that data (however my other points about standards is also relevant. 

L384 It is not clear if this is the retention index of the standard or the compound in the entrained extract. I would expect to see both in this table (as well as on a different column).

L391-405 This is all good

L408 I think you need to qualify this by indicating collections made in Kenya (or East Africa) because it is not necessarily the case in other regions of the world.

L414-416 this seems a key point that the Phlebotomus spp are found at all 3 sites. However it’s not entirely clear that that conclusion is supported by the evidence. The numbers are very low and it’s not clear from the Methods how much trapping effort ws deployed to achieve these cathes. Was it one night or 100 nights?

L435 “… like also found …” can this be expressed in a better way?

L453 Unfortunately I am not entirely convinced by this analysis. You could have tried some very simple manipulative experiments that would have relied on odor alone thereby vastly strengthening your conclusion. However as long as you clearly indicate that you understand the limitations of the study and discuss them the results have some validity. In my view you have tentatively identified some compounds that may act as attractants for the sand fly species in question. You have not shown that abundance and species distribution is linked to those compounds.

PLOS authors have the option to publish the peer review history of their article (what does this mean?). If published, this will include your full peer review and any attached files.

Reviewer #1: Yes: Joshua R. Lacsina

Reviewer #2: No
---

## [Decision Letter · Decision Letter 1]

25 Nov 2020

Dear Dr. Tchouassi,

Thank you very much for submitting your revised manuscript "Exploring the influence of different habitats and their volatile chemistry in modulating sand fly population structure in a leishmaniasis endemic foci, Kenya" for consideration at PLOS Neglected Tropical Diseases. As with all papers reviewed by the journal, your manuscript was reviewed by members of the editorial board and by several independent reviewers. The reviewers appreciated the attention to an important topic. Based on the reviews, we are likely to accept this manuscript for publication, providing that you modify the manuscript according to the review recommendations. 

As noted below, there remains one outstanding point for clarification regarding the identification of compounds on one column using co-injection with standards. All other requested revisions have been successfully addressed.

Sincerely,

Geraldine Marie Foster, PhD

Guest Editor

Helen Price

Deputy Editor

Reviewer's Responses to Questions

**Key Review Criteria Required for Acceptance?**

**Methods**

-Are the objectives of the study clearly articulated with a clear testable hypothesis stated?

-Is the study design appropriate to address the stated objectives?

-Is the population clearly described and appropriate for the hypothesis being tested?

-Is the sample size sufficient to ensure adequate power to address the hypothesis being tested?

-Were correct statistical analysis used to support conclusions?

-Are there concerns about ethical or regulatory requirements being met?

Reviewer #1: All critiques were addressed in revision, there are no methodological concerns.

Reviewer #2: (No Response)

**Results**

-Does the analysis presented match the analysis plan?

-Are the results clearly and completely presented?

-Are the figures (Tables, Images) of sufficient quality for clarity?

Reviewer #1: All critiques were addressed in revision, there are no concerns regarding the results.

Reviewer #2: (No Response)

**Conclusions**

-Are the conclusions supported by the data presented?

-Are the limitations of analysis clearly described?

-Do the authors discuss how these data can be helpful to advance our understanding of the topic under study?

-Is public health relevance addressed?

Reviewer #1: The conclusions have been appropriately revised to reflect the experimental findings -- there are no further concerns.

Reviewer #2: (No Response)

**Editorial and Data Presentation Modifications?**

Reviewer #1: Minor grammatical revisions will be needed before final publication.

Reviewer #2: (No Response)

**Summary and General Comments**

Reviewer #1: This is a revised manuscript by Hassaballa et al. reporting the results of a cross-sectional survey of sand fly species abundance and diversity in Rabai village of Baringo County, Kenya, as well as cataloguing profiles of volatile organic compounds (VOCs) in different habitats within the village. The characterization of VOCs as part of sand fly epidemiological studies is a novel approach and has the potential to be impactful for vector control efforts in East Africa. The authors have adequately addressed all the critiques I raised in my review. I commend the authors for this research, and recommend the revised manuscript be accepted for publication.

Reviewer #2: The authors have addressed the majority of the concerns that i raised major and minor. I apologies for confusing sessions with seasons and am happy with the meaning of both. My only outstanding issue is that you still can't categorically claim to identify compounds on one column by co-injection with standards.It doesn't detract from the manuscript/paper if they authors are clear about this issue and explain what they have done. But it will matter to those who read the paper and see that the authors are trying to claim full identification of compounds when they have not achieved that. However, this is a technical point and perhaps to most readers not important. But I will leave it to the authors to address this issue and "tone down" the identification rhetoric!

Still a nice piece of work well worthy of publication.

PLOS authors have the option to publish the peer review history of their article (what does this mean?). If published, this will include your full peer review and any attached files.

Reviewer #1: Yes: Joshua R. Lacsina

Reviewer #2: No
---

## [Editor Report · Decision Letter 2]

9 Dec 2020

Dear Dr. Tchouassi,

We are pleased to inform you that your manuscript 'Exploring the influence of different habitats and their volatile chemistry in modulating sand fly population structure in a leishmaniasis endemic foci, Kenya' has been provisionally accepted for publication in PLOS Neglected Tropical Diseases.

Best regards,

Geraldine Marie Foster, PhD

Guest Editor

Helen Price

Deputy Editor

---

## [Editor Report · Acceptance letter]

24 Jan 2021

Dear Dr. Tchouassi,

We are delighted to inform you that your manuscript, "Exploring the influence of different habitats and their volatile chemistry in modulating sand fly population structure in a leishmaniasis endemic foci, Kenya," has been formally accepted for publication in PLOS Neglected Tropical Diseases.

Best regards,

Shaden Kamhawi

co-Editor-in-Chief

Paul Brindley

co-Editor-in-Chief
